# The Influence of *CD28* Gene Polymorphism in Transfusion Reaction after Transfusing Leukoreduced Blood Components

**DOI:** 10.3390/jcm9030877

**Published:** 2020-03-23

**Authors:** Ding-Ping Chen, Wei-Tzu Lin, Wei-Ting Wang, Tzong-Shi Chiueh

**Affiliations:** 1Department of Laboratory Medicine, Linkou Chang Gung Memorial Hospital, Taoyuan 33305, Taiwan; berry0908@cgmh.org.tw (W.-T.L.); s1223@adm.cgmh.org.tw (W.-T.W.); drche0523@cloud.cgmh.org.tw (T.-S.C.); 2Medical Biotechnology and Laboratory Science, Chang Gung University, Taoyuan 33302, Taiwan; 3Graduate Institute of Biomedical Sciences, College of Medicine, Chang Gung University, Taoyuan 33302, Taiwan

**Keywords:** transfusion reaction (TR), CD28, gene polymorphism

## Abstract

CTLA-4 and CD28 belong to co-stimulation molecules, the abnormal expression of which can regulate the T cell activation and then affect the degree of immune response. Moreover, blood transfusion reaction (TR) is a kind of immune reaction. Consequently, the hypothesis of this study was that the TR still occurred after transfusing leukoreduced blood components as a result of the sensitivity of immune system, and a small number of immune stimulations were enough to induce an immune response in patients. There were 38 cases and 36 healthy controls included in this study. The selected *CD28* gene were sequenced to search single nucleotide polymorphism (SNPs), and the correlation between TR and these SNPs was analyzed. According to our data, only the rs3181097 (promoter, −1059) of *CD28* gene polymorphism was associated with TR. The *p* value of rs3181097 under the co-dominant model was 0.031. GG was used as a reference genotype, the *p* value of GG vs. AG was 0.010 (OR: 0.210, 95% CI: 0.062–0.712), and GG vs. AG + AA was 0.028 (OR: 0.292, 95% CI: 0.095–0.901). In addition to *CTLA-4*, *CD28* gene was also a crucial SNP in TR, because there was a signification for the haplotype with G_rs3181097_ (*p* = 0.015). Consequently, we suggested that the TR was related to *CD28* by regulating the degree of immune response.

## 1. Introduction

Transfusion reaction (TR) is a kind of immune reaction, which is defined as the adverse reaction after transfusion, including of fever, allergy, dyspnea, and so on [1]. There are many causes for TR, which may be induced by red blood cell antigen (ABO antigen), white blood cell antigen (HLA antigen), platelet antigen (HPA antigen) or cytokines (IL-1, IL-6, and so on) [2,3]. When heteroantigen is first recognized by immune system, the antigen-specific antibody will be produced rapidly against the same antigen, as it is recognized again. Therefore, TRs are more common in patients with multiple transfusions [4] and in multiparous women [5]. The TR can develop even after removal of these possible factors by crossmatching and using pre-storage leukocyte-reduced components before infusing. It might be caused by other host factors in blood components, and a few molecules are enough to trigger an immune response.

The effectiveness of hematopoietic stem cell transplantation (HSCT) could be affected by the abnormal expression of CTLA-4 and CD28 [6]. Moreover, hematopoietic stem cells (HSCs) have an ability to differentiate into all the blood cell types [7]. In other words, the components in the blood product are mature cells that are differentiated from HSCs. In addition, both transplantation and blood transfusion inject allogeneic cells into patients. Therefore, we speculated that the mechanism of TR and transplantation rejection was similar. Furthermore, CTLA-4 and CD28 belong to co-stimulation molecules, which regulate the T cell activation and then affect the degree of immune response [8]. Consequently, we thought the expression of CTLA-4 and CD28 should also be related to TR. The hypothesis of this study was that TR still occurred after transfusing leukoreduced blood components as a result of the excessive immune response in patients, and a small number of stimulations could trigger an immune response.

Our previous study indicated that *CTLA-4* gene polymorphism was associated with TR after the infusion of leukoreduced blood components [9]. Furthermore, the common points of *CD28* and *CTLA-4* genes are located in the 2q33, as immune regulator, mutually homologous, and have the same ligands (B7 family) [10]. T cell homeostasis was influenced by an imbalanced expression level of these two molecules [11]. Moreover, there were several autoimmune diseases and cancers simultaneously associated with *CTLA-4* and *CD28* genes. For example, type 1 diabetes was related to both the *CD28* gene [12] and the *CTLA-4* gene [13], as well as rheumatoid arthritis [14,15] and cancer risk in Asians [16,17]. Consequently, we conjectured that *CD28* might be a genetic risk factor of alloimmunization and associated with TR, besides *CTLA-4*.

Comparing CD28 and CTLA-4, CD28 is continuously expressed on the surface of naïve and activated T cells [18], which functions to promote T cell activation and increase antibody production of B lymphocytes [19,20]. CD28-deficient mice were unable to activate T cells even when stimulated by antigens and anti-CD3 monoclonal antibodies [21] and resulted in T lymphocyte tolerance and anergy [22]. CTLA-4 is continuously expressed on the surface of regulatory T cells (Treg), and its expression on activated T cell is induced by CD28 and the T cell receptor signal [23], which functions to inhibit T cell activation [24]. CTLA-4-deficient mice might develop large-scale lymphoproliferative diseases or spontaneous autoimmune diseases, and die in 4–5 weeks [25,26].

In addition, the fetal blood enters into the maternal circulation and they achieve mutual tolerance and immune balance during pregnancy [27]. When this immune balance was destroyed, it would cause spontaneous abortion [28]. One of the reasons for spontaneous abortion was regulated by immune-related genes [29]. There were several studies indicating that spontaneous abortion was related to *CTLA-4* and *CD28* polymorphisms and serum level [30,31,32,33]. To protect the fetus during pregnancy, maternal alloimmunity reactions would be inhibited by the mother’s own Treg in opposition to paternal antigens among embryo cells [34]. Moreover, CD28 was the major factor to promote the proliferation of Treg [11]. Additionally, TR is also an immune response caused by allogeneic blood, which supported us to infer that TR was also related to *CD28*. Therefore, the aim of the study was to identify the relationship between *CD28* genotype and TR and further explore the mechanism of blood TR caused by gene mutation.

## 2. Experimental Section

### 2.1. Study Subjects

The study was approved by the Institutional Review Board of Chang Gung Memorial Hospital (CGMH) with the approval ID of 201901246B0. There were 74 participants included in this study at Linkou CGMH from March 2019 to January 2020. The 38 patients were divided into test groups, who had TR after infusing leukoreduced blood products (leukocyte-poor red blood cell, LPR, or leukocyte-poor platelet, LPP). The patients with cancer or autoimmune disease or immune suppression drugs or the use of steroids were not excluded. The diagnostic criteria of TRs were set according to the Hemovigilance Module Surveillance Protocol of National Healthcare Safety Network (NHSN) Biovigilance Component [35]. Another 36 participants were healthy volunteers without autoimmune disease and cancer.

### 2.2. DNA Extraction

The genomic DNA was extracted from peripheral blood of patients and health controls, and a QIAamp DNA Mini kit (Qiagen GmbH, Hilden, Germany) was used according to the manufacturer’s instructions. DNA concentration was assessed through a UV spectrometer, and the purity of DNA was evaluated by measuring the optical density at 260 and 280 nm.

### 2.3. PCR Amplification

Analysis of the correlation between TR and SNPs of *CD28* was performed in the Department of Laboratory Medicine at Linkou CGMH. The PCR primers used in this study are shown in Table 1. We only focused on the promoter and intron 3 region of CD28, because we wanted to investigate whether the expression level of CD28 was associated with TR, and rs3116496 in the intron 3 region of *CD28* was a hotspot in literature. The 25 μL of PCR reaction volume contained 1 μL of DNA, 10 μL of HotStarTaq DNA Polymerase (Qiagen GmbH, Hilden, Germany), 1 μL of 10 μM forward primer, 1 μL of 10 μM reverse primer, and 12 μL of ddH_2_O. The PCR program was 95 °C for 3 min at initial denaturation step and 30 cycles of 95 °C for 30 sec, 58 °C for 30 sec, and 72 °C for 2 min. The final extension step was 10 min at 72 °C and finally soaking at 10 °C indefinitely. For gel electrophoresis visualization, 5 μL of the PCR products was pipetted onto a 1.5% agarose gel and conducted at 100 V for 30 min, and then subjected to UV illumination to ensure the correctness of PCR products.

### 2.4. Purification and SNP Analysis

The enzyme containing 2.5 µL shrimp alkaline phosphatase and 0.05 µL exonuclease I (New England Biolabs, UK) was used to purify PCR products, and then the PCR products were sequenced by using ABI PRISM 3730 DNA analyzer (Applied Biosystems, Foster City, CA, USA). The SNP analysis was performed on the promoter and intron 3 region of *CD28*, wherein rs1879877, rs3181096, rs3181097, rs3181098, rs56228674, and rs3116496 were selected for genotyping. The data of SNP genotyping were not all available because of unclear sequencing information.

### 2.5. Statistical Analysis

The Hardy–Weinberg equilibrium (HWE) was conducted for the control group to reduce sampling bias in case–control studies. The statistical correlation of allele frequency, genotype frequency, and haplotype was analyzed by SPSS 17.0 through chi-square test and Fisher’s exact test. Haplotype analysis included four SNPs of the *CTLA-4* gene (rs4553808, rs62182595, rs16840252, and rs5742909), which were related to TR [9], and the meaningful *CD28* SNP in this study.

## 3. Results

The characteristics of participants are shown in Table 2. The average age of patients was 54.3 years old, and the gender ratio was 27:11, in which women accounted for 71%. A febrile non-hemolytic transfusion reaction (FNHTR) (67.8%) constituted the majority of TR in the study. Furthermore, there were 17 of 38 patients transfused with LPR, and the others were transfused with LPP. In which, there were two patients who simultaneously had an allergic reaction and FNHTR in a transfusion. The average age of the control group was 22.5 years, and women accounted for 62%.

The selected *CD28* gene SNPs (rs1879877, rs3181096, rs3181097, rs3181098, and rs3116496) were genotyped in patients and healthy controls. All SNPs complied with HWE, *p* > 0.05, and there was no SNP showing differences in allele frequency between patients with TR and healthy controls (Table 3).

All genotypes of *CD28* were not significantly related to TR, except rs3181097. The *p* value of rs3181097 under co-dominant model (GG vs. AG vs. AA) was 0.031. GG was used as a reference genotype, and the *p* value of GG vs. AG was 0.010 (OR: 0.210, 95% CI: 0.062–0.712), and GG vs. AG + AA was 0.028 (OR: 0.292, 95% CI: 0.095–0.901). The complete data are shown in Table 4.

Haplotype analysis was conducted with blood transfusion-related SNPs of *CTLA-4* gene in our previous study [9] and the statistically significant SNP of *CD28* in this study (Table 5). The *p* value was 0.007 when the haplotype contained only those four *CTLA-4* SNPs (OR: 4.571, 95% CI: 1.417–14.748). Adding the SNP of *CD28* into the haplotype, it was statistically significance when rs3191097 had a G allele (*p* = 0.015, OR: 2.591, 95% CI: 1.197–5.606). However, it was not associated with TR when rs3181097 had an A allele, *p* = 0.718.

## 4. Discussion

The co-stimulation pathway is a central factor for inducing effective antigen-specific immune response [36]. Leukoreduced components will be used when the patient has had multiple transfusions or TR before. Moreover, multiple transfusions may produce antibodies other than anti-ABO antibodies and anti-HLA antibodies. Our results showed that most of the TR were FNHTR. The main causes of FNHTR are HLA antigen and cytokines released by leukocytes [37]. The number of leukocytes <5 × 10^6^/unit could effectively prevent the generation of alloimmunity [38]. However, some cases have FNHTR even after reducing the number of leukocytes and cytokines by using pre-storage leukoreduced components [39]. That might be explained by the leukoreduced blood products still containing a small number of leukocytes and cytokines [40]. The number of leukocytes in LPR was about 1 × 10^5^/unit after filtering by a Cellselect filter and the number of leukocytes in LPP was about 0.4 × 10^5^/unit after filtering by a cotton-wool (Imugard) filter [40]. Therefore, it was indicated that TRs after transfusing leukoreduced components were related to the sensitivity of immune system, and a small number of immune-stimulations were enough to trigger an immune response in the patient [41].

Combining with the results of a previous study [9], *p* value was 0.007 (OR: 4.571, 95% CI: 1.417–14.748) when the haplotype contained only *CTLA-4* SNPs (A_rs4553808_G_rs62182595_Cr_s16840252_C_rs5742909_). While adding *CD28* SNP into this haplotype, the *p* value was significantly statistical different when allele of rs3181097 was G (*p* = 0.015), but A was not (*p* = 0.718). Therefore, it was represented that the rs3181097 of *CD28* gene also was a crucial SNP in TR.

The SNP of rs4553808 (-1661) in the *CTLA-4* promoter region would affect the transcriptional binding activity for CCAAT/enhancer-binding protein beta (c/EBPβ) and nuclear factor I and influence the transcription level of the *CTLA-4* gene [42], and the SNP of rs5742909(-318) was associated with the expression of CTLA-4 [43] too. However, rs16840252 (-1146) could be affected by rs4553808 or rs5742909 of *CTLA-4* gene, because there was a strong linkage disequilibrium between them [44]. The SNP of rs3181097 (-1059) in the promoter region of *CD28* was related to the clinical pathology of childhood IgA nephropathy in Korean ethnic groups [44]. In the position of rs3181097, specificity protein 1 (SP1) and c/EBPα transcription factors could bind to the A-containing sequence, but only SP1 could bind to the G-containing sequence [45]. Consequently, the SNP of rs3181097 would affect the transcription factor binding sites and directly or indirectly changing the expression of CD28 and regulated T cell-mediated immune response [46] then resulting in TR.

These SNPs (rs4553808, rs62182595, rs16840252, rs5742909, and rs3181097) are all in the promoter region of the gene, and gene variation in the promoter region would affect the expression level [47]. Hence, the mechanism of TR caused by these SNPs might result from the expression level of CTLA-4 and CD28. However, the function of rs62182595 and rs3181097 SNP has not been investigated yet. Consequently, the mechanism of TR caused by rs3181097 was not affirmed. Moreover, the blood product contained more than one factor that could result in TR and the patient’s physical condition was different, so the TR was regulated by multiple factors.

## 5. Conclusions

The gene polymorphisms of *CD28* and *CTLA-4* were both associated with TR, and it might be due to the regulation of T cell activation and immune response. In the future, the effect of the SNP on CD28 and CTLA-4 expression should be further investigated by cell model study, and the biochemical pathways of TR caused by the expression of CD28 and CTLA-4 should also be investigated. These efforts could effectively prevent the occurrence of TR and improve the safety of blood transfusion. However, the limitation of this study was that the patients with cancer, autoimmune disease, or drug used were not exclude.

## Figures and Tables

**Table 1 jcm-09-00877-t001:** The CD28 primers used in this study. F: forward primer; R: reverse primer.

Sequence	GC Count	Tm	bp
Promoter			
F: 5′- GGG TGG TAA GAA TGT GGA TGA ATC-3′	24/11 46%	63.6 °C	1542
R: 5′-CAA GGC ATC CTG ACT GCA GCA-3′	21/12 57%	63.2 °C	
Intron3			
F: 5′- AAG GAT GCA GTT TAG GGT CTA GAT T -3′	25/10 40%	62.5 °C	886
R: 5′-GAT CAA GCC AAC ATT GTC CAT TGG-3′	24/11 46%	63.6 °C	

GC-content was defined as count (G + C)/ count (A + T); Tm was defined as the temperature at half the time the double helix structure of DNA degrades; bp: base pairs.

**Table 2 jcm-09-00877-t002:** Characteristics of patients (*n* = 38) with transfusion reaction (TR) after transfusing leukoreduced blood components and healthy controls (*n* = 36).

	Patients, No. (%)	Controls, No. (%)
Average age of patients	54.3 ± 3	22.5 ± 0.96
Gender (female)	71	62
Type of blood transfusion		
Leukocyte-poor platelet	17	
Leukocyte-poor red blood cell	21	
Type of transfusion reaction		
Allergic reaction	15 (39.5)	
Febrile non-hemolytic transfusion reaction (FNHTR)	25 (65.8)	
Both allergic reaction and FNHTR	2 (5.3)	

**Table 3 jcm-09-00877-t003:** Hardy–Weinberg equilibrium (HWE) in control group, allele frequencies and odds ratio for TR in patients and controls.

SNP	Position	Allele	Minor Allele Frequency		HWE *p* Value	Odds Ratio 95% CI	*p*^a^ Value
			Patient	Control			
rs1879877	203705277	G/T	0.394	0.347	0.889	1.226 (0.628–2.393)	0.550
	(−1198)						
rs3181096	203705369	C/T	0.276	0.208	0.907	1.582 (0.732–3.417)	0.241
	(−1106)						
rs3181097	203705416	G/A	0.400	0.500	0.230	0.667 (0.330–1.345)	0.257
	(−1059)						
rs3181098	203705655	G/A	0.257	0.208	0.907	1.313 (0.607–2.840)	0.489
	(−820)						
rs3116496	203729789	T/C	0.157	0.143	0.615	1.156 (0.457–2.923)	0.759
	(IVS3 +17)						

SNP: single nucleotide polymorphism; HWE: Hardy–Weinberg equilibrium; 95 % CI: 95% confidence interval; *p*^a^ values were counted from chi-square test or Fisher’s exact test.

**Table 4 jcm-09-00877-t004:** Genotypes of *CD28* SNPs and their correlations with risk of TR.

SNP	Genotype	OR (95% CI)	*p* Value
Promoter			
rs1879877	GG vs. GT vs. TT		0.714
	GG	ref.	ref.
	GT	1.715 (0.452–6.506)	0.426
	TT	1.600 (0.430–5.958)	0.482
	GT + TT	1.653 (0.486–5.628)	0.418
rs3181096	CC vs. CT vs. TT		0.647
	CC	ref.	ref.
	CT	0.773 (0.285–2.094)	0.612
	TT	0.457 (0.076–2.755)	0.667
	CT + TT	0.698 (0.275–1.776)	0.450
rs3181097	GG vs. AG vs. AA		**0.031**
	GG	ref.	ref.
	AG	0.210 (0.062–0.712)	**0.010**
	AA	0.519 (0.134–2.018)	0.341
	AG + AA	0.292 (0.095–0.901)	**0.028**
rs3181098	GG vs. AG vs. AA		0.651
	GG	ref.	ref.
	AG	0.957 (0.345–2.652)	0.932
	AA	0.478 (0.079–2.879)	0.668
	AG + AA	0.829 (0.322–2.133)	0.697
**Intron3**			
rs3116496	TT vs. CT vs. CC		0.601
	TT	ref.	ref.
	CT	0.987 (0.356–2.708)	0.972
	CC	NA	0.491
	CT + CC	1.080 (0.398–2.927)	0.880

Bold type in *p* value means that it had significant difference between patients and healthy controls, *p* < 0.05. OR: odds ratio; CI: confidence interval.

**Table 5 jcm-09-00877-t005:** The haplotype analysis of *CTLA-4* (rs4553808, rs62182595, rs16840252, rs5742909) and *CD28* (rs3181097) gene SNPs.

Haplotype	OR (95% CI)	*p* Value
Only *CTLA-4*		
AGCC	4.571 (1.417–14.748)	**0.007**
Combine *CTLA-4* and *CD28*		
AGCCG	2.591 (1.197–5.606)	**0.015**
AGCCA	0.866 (0.397–1.889)	0.718

OR: odds ratio, CI.: confidence interval. Bold type: significant difference between patients and healthy controls, *p* < 0.05.

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
