# Peer review of "The Influence of CD28 Gene Polymorphism in Transfusion Reaction after Transfusing Leukoreduced Blood Components"

_jcm, 2020, doi:10.3390/jcm9030877_

Round 1

Reviewer 1 Report

Firstly the manuscript needs extensive English revision: there are multiple grammatical errors.

Regarding the content, the Abstract and the Introduction need to clarify what CD28 and CTLA-4 represent.

The Experimental Section is better but again needs more clarification as to what constitutes the "test group". I assume it is the group that had transfusion reactions but this needs to be stated in 2.1.

In the Results section it would be nice if the Test Group was compared to the "Control Group" (Table 2). 

In the Discussion section there is a problem with exponents e.g. 0.4 x105/unit should be 0.4 x 10^4/unit. Again the apart from grammatical errors the discussion and conclusion need more clarity. I assume the idea was to state that transfusion reactions are more common in one or two specific polymorphisms but that is not clear.

Author Response

  1. Firstly the manuscript needs extensive English revision: there are multiple grammatical errors.

Response: We appreciated the reviewer’s comment, and we had rigorously revised this manuscript.

  1. Regarding the content, the Abstract and the Introduction need to clarify what CD28 and CTLA-4 represent.

Response: Thanks for the Reviewer’s comment. We had added the description into revised manuscript to clarify the role of CTLA-4 and CD28 in transfusion reaction. (row 11 - 15, row 43 – 48)

  1. The Experimental Section is better but again needs more clarification as to what constitutes the "test group". I assume it is the group that had transfusion reactions but this needs to be stated in 2.1.

Response: Thanks for the Reviewer’s comment. The description for the constitution of test group had been stated in 2.1. (row 81 – 86)

  1. In the Results section it would be nice if the Test Group was compared to the "Control Group" (Table 2).

Response: Thanks for the Reviewer’s comment. We had added one column into Table 2 to show the data of controls.

  1. In the Discussion section there is a problem with exponents e.g. 0.4 x105/unit should be 0.4 x 10^5/unit. Again the apart from grammatical errors the discussion and conclusion need more clarity. I assume the idea was to state that transfusion reactions are more common in one or two specific polymorphisms but that is not clear.

Response: Thanks for the Reviewer’s comment. We had added the “^” to indicate exponent. (row 153, 157, and 158)

Reviewer 2 Report

  1. Introduction should be revised. Two short sentences about physiological function of CD28 and CTLA-4 should be added and some details about the mechanisms by which CD28 and CTLA-4 regulate T cell homeostasis should be discussed.
  2. Row 58: please add "polymorphisms and serum level" at the end of the sentence.
  3. The part concerning the analysis of correlation between TR and SNPs of CD28 mentioned from the row 72 to the row 76 should be discussed in a distinct paragraph from "Study subjects".  
  4. Row 70: please specify if the 74 participant were selected according to some inclusion/exclusion criteria.
  5. Table2: baseline characteristics of both groups should be compared in order to highlight significant differences
  6. Table2: "type of transfusion-associated" should be changed in "type of transfusion reaction" or "type of TR"
  7. Table2: the table should be re-edited preferring left alignment
  8. Table2: please specify how many patients had both allergic reaction and FNHRT 
  9. Row 125: I think that "AG+GG" should be replaced by "AG+AA", please verify. 
  10. Table 4: for rs3116496 the first row should be "TT vs CT vs CC" instead, please verify. 
  11. Row 141: 5x106 should be replaced by 5x106. The same mistake has been repeated for all the other number. 
  12. Row 146 and 151: missing reference.
  13. Row 166: the author should try to explain the hypothetical mechanism by which the difference binding ability of CEBPA and SP1 to CD28 A-containing sequence vs CD28 G-containing sequence affect the T cell response and transfusion reaction 
  14. An extensive English editing is required given that some sentences are too difficult to understand. 
  15. Through the manuscript abbreviations are sometimes missing (e.g. tranfusion reaction in row 70 should be abbreviated in TR; FNHTR is not specify in its first mention in row 17). Please revised the entire manuscript to fix them. 

Author Response

  1. Introduction should be revised. Two short sentences about physiological function of CD28 and CTLA-4 should be added and some details about the mechanisms by which CD28 and CTLA-4 regulate T cell homeostasis should be discussed.

Response 1: Thanks for the Reviewer’s comment. We had added the description into abstract and the introduction to clarify the role of CTLA-4 and CD28 in transfusion reaction. (row 11 - 15, row 43 – 48)

  1. Row 58: please add "polymorphisms and serum level" at the end of the sentence.

Response 2: Thanks for the Reviewer’s comment. We had added "polymorphisms and serum level" at the end of the sentence. (row 71)

  1. The part concerning the analysis of correlation between TR and SNPs of CD28 mentioned from the row 72 to the row 76 should be discussed in a distinct paragraph from "Study subjects".

Response 3: Thanks for the Reviewer’s comment. We had removed the sentence for analysis of correlation between TR and SNPs of CD28 from 2.1 Study subjects to 2.3 PCR Amplification.

  1. Row 70: please specify if the 74 participants were selected according to some inclusion/exclusion criteria.

Response 4: Thanks for the Reviewer’s comment. These descriptions were added into revised manuscript. (row 81 – 87)

  1. Table2: baseline characteristics of both groups should be compared in order to highlight significant differences

Response 5: Thanks for the Reviewer’s comment. We had added one column into Table 2 to show the data of controls.

  1. Table2: "type of transfusion-associated" should be changed in "type of transfusion reaction" or "type of TR"

Response 6: Thanks for the Reviewer’s comment. It had been edited in accordance with the reviewer comments.

  1. Table2: the table should be re-edited preferring left alignment

Response 7: Thanks for the Reviewer’s comment. It had been edited in accordance with the reviewer comments.

  1. Table2: please specify how many patients had both allergic reaction and FNHRT

Response 8: Thanks for the Reviewer’s comment. According to our data, two patients had both allergic reaction and FNHRT in a transfusion, and which had been added into Table 2 and row 125 - 126.

  1. Row 125: I think that "AG+GG" should be replaced by "AG+AA", please verify.

Response 9: Thanks for the Reviewer’s reminder. It had been edited to AG+AA. (row 136)

  1. Table 4: for rs3116496 the first row should be "TT vs CT vs CC" instead, please verify.

Response 10: Thanks for the Reviewer’s reminder. It had been edited to TT vs CT vs CC in Table 4.

  1. Row 141: 5x106 should be replaced by 5x106. The same mistake has been repeated for all the other number.

Response 11: Thanks for the Reviewer’s comment. We had added the “^” to indicate exponent. (row 153, 157, and 158)

  1. Row 146 and 151: missing reference.

Response 12: Thanks for the Reviewer’s comment. The reference had been added in row 154 and row 158.

  1. Row 166: the author should try to explain the hypothetical mechanism by which the difference binding ability of CEBPA and SP1 to CD28 A-containing sequence vs CD28 G-containing sequence affect the T cell response and transfusion reaction

Response 13: Thanks for the Reviewer’s comment. The hypothetical mechanism of rs3181097 polymorphism in transfusion reaction had been explained in revised manuscript. (row 170 – 176)

  1. An extensive English editing is required given that some sentences are too difficult to understand.

Response 14: We appreciated the reviewer’s comment, and we had rigorously revised this manuscript.

  1. Through the manuscript abbreviations are sometimes missing (e.g. tranfusion reaction in row 70 should be abbreviated in TR; FNHTR is not specify in its first mention in row 17). Please revised the entire manuscript to fix them.

Response 15: Thanks for the Reviewer’s comment. The abbreviations in the entire manuscript had been rechecked.

Round 2

Reviewer 2 Report

I really appreciates the efforts of the authors to improve the quality of this work.

Despite an English revision was clearly performed, some further corrections are needed.

For example, the first sentence of the abstract isn't correct, I suggest to replace it with the following: "CTLA-4 and CD28 belong to co-stimulation molecules and their abnormal expression regulates T cell activation, thus affecting the degree of immune response". 

Other grammatically incorrect sentences are present through the manuscripts.

Please be sure that an author with proved English experience revise the work.